# Fission and fusion scenarios for magnetic microswimmer clusters

Francisca Guzmán-Lastra[1], Andreas Kaiser[2] & Hartmut Löwen[1]

Fission and fusion processes of particle clusters occur in many areas of physics and chemistry from subnuclear to astronomic length scales. Here we study fission and fusion of magnetic microswimmer clusters as governed by their hydrodynamic and dipolar interactions. Rich scenarios are found that depend crucially on whether the swimmer is a pusher or a puller. In particular a linear magnetic chain of pullers is stable while a pusher chain shows a cascade of fission (or disassembly) processes as the self-propulsion velocity is increased. Contrarily, magnetic ring clusters show fission for any type of swimmer. Moreover, we find a plethora of possible fusion (or assembly) scenarios if a single swimmer collides with a ringlike cluster and two rings spontaneously collide. Our predictions are obtained by computer simulations and verifiable in experiments on active colloidal Janus particles and magnetotactic bacteria.

[1] Institut für Theoretische Physik II: Weiche Materie, Heinrich-Heine-Universität Düsseldorf, D-40225 Düsseldorf, Germany. [2] Materials Science Division, Argonne National Laboratory, 9700 S Cass Avenue, Argonne, Illinois 60439, USA. Correspondence and requests for materials should be addressed to A.K. (email: akaiser@anl.gov) or to H.L. (email: hlowen@thphy.uni-duesseldorf.de).

Many phenomena in physics, chemistry and biology are governed by fission and fusion of particle clusters. These processes occur on widely different length and energy scales ranging from subnuclear to astronomic. Examples include the large hadron collider[1], nuclear fission and fusion[2], splitting and merging of atomic and molecular clusters[3–5], micron-size colloidal particles[6,7], cells[8] and macroscopic granulates[9] up to clustering in planetary debris disks[10]. Fission and fusion can either happen spontaneously or be induced by a collision with another particle or entity. The fission and fusion processes provide valuable insight into the interactions and dynamics of the individual particles and are a promising pathway to fabricate new types of composite matter on the various scales.

In the past decade microswimmers have been studied extensively. These active soft matter systems consume energy and are, therefore, self-propelled out-of-equilibrium autonomous systems. The interactions between many of those swimmers are either direct body forces or hydrodynamic ones as mediated by the solvent flow field created by the swimming motion. The latter depend on the details of the swimming mechanism and can be classified as either neutral, pushers or pullers[11,12]. An important example for the former are magnetic dipole moments that enable to actuate and control the swimming path by an external magnetic field[13–20]. In the absence of swimming, the structure of two-dimensional[21,22] and three-dimensional[23,24] dipole clusters has been explored recently, but the impact of activity on a swimmer cluster has been rarely studied[25]. Recent simulation studies[26,27] have revealed that the mean cluster size crucially depends on the strength of the self-propulsion speed. Moreover, it has been shown that activity itself[28–31] and chemotaxis[32,33] can induce clustering[28–31]. Hence, though the structure of microswimmer clusters have been widely studied, the process of induced and spontaneous fusion of such clusters is unexplored except for spontaneous fission of swimmers clusters where all hydrodynamics was neglected. Furthermore, the impact of hydrodynamic interactions (HIs) has not yet been studied for magnetic dipole swimmers.

In this communication, we explore both spontaneous and induced fission and fusion processes for magnetic microswimmer clusters. These processes may also be referred to as disassembly and assembly. We do this for the two basic swimmer classes pushers and pullers. The details of the HIs governing whether the swimmer are pushers, neutral or pullers[12,34–36]. Pushers and pullers can be described as dipole swimmers whose flow fields are similar, but with opposite flow patterns. While pushers push fluid away from the body along their swimming axis and draw fluid in to the sides, pullers pull fluid in along the swimming direction and repel fluid from the sides. This has important consequences for the interactions between swimmers and give rise to qualitatively different fission and fusion scenarios, which we classify for small particle clusters. In detail, we study the spontaneous fission processes for two chosen types of clusters, linear chains and rings, representing the ground state of magnetic dipoles[21,24]. While linear chains remain stable for pullers and neutral swimmers they are split in the case of pushers. However, ring-like clusters show fission events in any case independent on the details of the HI. A single swimmer hitting a ring cluster can end up in spontaneous and induced fission or in fusion. Finally, two fusing ring clusters reveal a wealth of complex joint dynamical modes. We use computer simulation techniques to obtain our predictions, which are in principle verifiable in experiments on active colloidal Janus particles and magnetotactic bacteria.

## Results

**Spontaneous fission of a chain.** Let us start using the simplest structure of a cluster—a one dimensional chain of $N$ dipolar

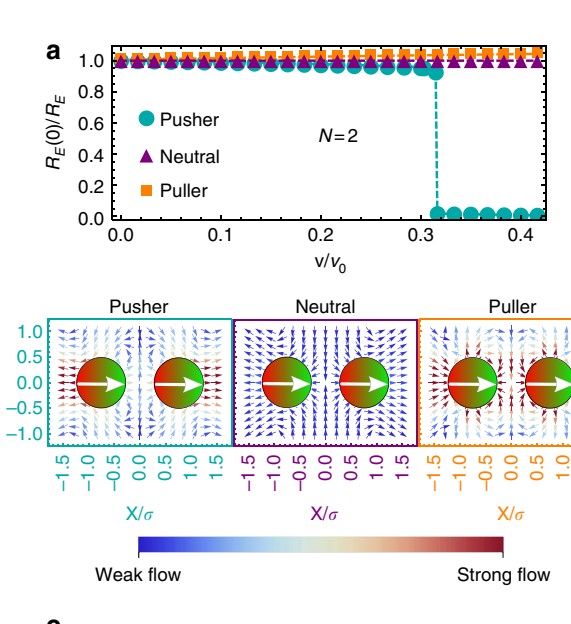

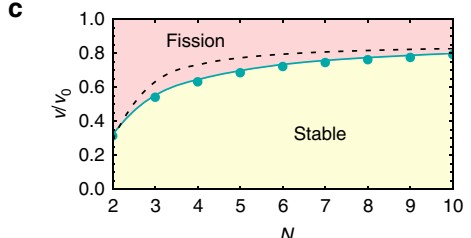

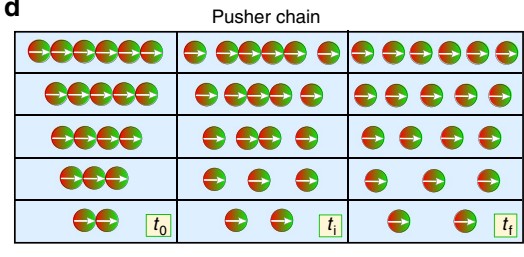

**Figure 1 | Stability of microswimmer chains.** (**a**) Relative end-to-end distance $R_E$ for all three HI cases and varied velocity $v$ for fixed $N = 2$. (**b**) Flow fields for a chain of $N = 2$ magnetic microswimmers for (left) pushers, (middle) neutral swimmers and (right) pullers. The magnitude of the flow is indicated by the colour coding of the arrows, weak flow is indicated by dark blue and strong flow by dark red arrows. (**c**) Emerging state diagram spanned by reduced velocity $v/v_0$ and number $N$ of particles within a chain of pushers marking two distinct regions; stable chains for small and fission for large self-propulsion velocities. Numerical results (cyan line) obtained for soft sphere dipoles are compared with those for hard sphere dipoles (black dashed line). (**d**) Time series indicating the fission process for the cases $N = 2, \ldots, 6$, showing the initial configurations at time $t_0$, the configuration after the first fission event at an intermediate time $t_i$ and the final steady state configuration of $N$ individual particles at time $t_f$, see also Supplementary Movies 1 and 2.

particles with a fixed dipole moment **m**, where all magnetic moments possess the same orientational direction aligned along their separation vector. So the total magnetic moment $M = \left| \sum_{i=1}^{N} \mathbf{m}_i \right|$ is $M = Nm$ for all chains, which coincides with the ground state for small numbers of particles[24]. By applying a self-propulsion velocity **v**, directed along the dipole moments, to each single dipolar particle, a joint motion of the linear chain will emerge. Using the inverse of the end-to-end distance

$$R_E = |\mathbf{r}_N - \mathbf{r}_1|, \qquad (1)$$

where $\mathbf{r}_i = [x_i, y_i, z_i]$ is the coordinate of the $i$th particle, we study the spatial extension of the chain for varied self-propulsion velocity $v$, see Fig. 1a for the case of $N = 2$, whereby, $R_E(0)$ depicts the value for a passive chain and $R_E$ is measured after a sufficiently long simulation time $t$ to establish steady-state structures. While in the absence of HIs no changes in the chain conformation are observed[37], we identify three different behaviours for magnetic microswimmers. (i) Pullers, generating a contractile flow field, shrink the chain, $R_E(0)/R_E > 1$, (ii) neutral swimmers hardly affect the configuration of the chain, $R_E(0)/R_E \geq 1$, and (iii) for pushers, generating an extensile flow field, the chain swells, $R_E(0)/R_E < 1$. This even leads to fission of the chain $R_E(0)/R_E \rightarrow 0$, for reduced self-propulsion velocities $v/v_0 \geq 0.32$, where $v_0 = 2m^2/\pi\eta\sigma^5$ is the corresponding drag velocity caused by the magnetic dipole–dipole interaction for two aligned and touching dipoles, with diameter $\sigma$, in a head-to-total configuration. This observed behaviour is the result of the super-positioned flow field of the microswimmers, which is contractile for pullers and extensile for pushers, see Fig. 1b.

The state diagram for chains of pushers spanned by the applied self-propulsion velocity $v$ and the number of microswimmers $N$ within a chain is shown in Fig. 1c. This diagram reveals two distinct regimes. For low velocities the chain remains stable, while large self-propulsion velocities lead to a fission. The fission process for chains of pushers happens stepwise as can be seen in the time series sketched in Fig. 1d and in the Supplementary Movies 1 and 2. The extensile flow field generated by pushers leads to a repulsion of the particles within the chain. Beyond a certain critical velocity, the chain of $N$ particles will split into three different units—a remaining chain of $N - 2$ pushers and repelled head and tail particle. To understand the symmetric stepwise mechanism we will address below the behaviour of the critical velocity leading to fission, as well as an analysis of all acting forces within a chain.

Let us now focus on the behaviour of the critical velocity leading to fission as a function of number of swimmers before we discuss the fission process itself. We have already shown that the extensile flow of the pushers can lead to fission although the particles are attracted due to their dipole–dipole interaction. By considering the total acting magnetic force on the head particle of a chain $F_1$, we can observe that the magnitude of the magnetic attraction grows with increasing chain length $N$. Here we compare the cases of soft magnetic dipolar particles with hard sphere dipoles. In Fig. 2, the force on the head particle $F_1$ is normalized by $F_0 = 6m^2/\sigma^4$, where $\sigma$ is the respective length scale provided by the hard core potential or the used Weeks-Chandler-Anderson (WCA) potential[38]. The force $F_0$ is the force due to the dipole–dipole interaction if a particle pair is in a head-to-tail configuration with a central distance $\sigma$. Hence for a larger number of pushers in the chain higher self-propulsion velocities are necessary to split the chain. As a remark, we add these data to Fig. 1c as a dashed line, leading again to differences that are caused by the softness of the pair potential.

To determine why the chain splits in the middle, we focus on all acting forces on the microswimmers within a chain. Due to symmetry, torques are excluded for all chains and for the middle microswimmer in a chain with an uneven number of swimmers, the only unbalanced force is the force due to self-propulsion, see Fig. 3a. The two ends are attracted by the dipole–dipole interaction and repelled by the Stokeslet. While for pullers there is a further attraction of the particles in the end of the chain, the extensile flow field generated by pushers lead to a further repulsion—which is proportional to the self-propulsion velocity $v$. Hence, beyond a critical velocity $v_c$ the chain of $N \geq 3$ particles will split into three distinct units, two individual swimmers and a remaining chain of $N - 2$ swimmers, see again Fig. 1d.

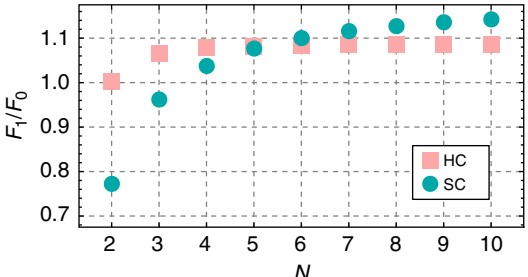

**Figure 2 | Magnetic dipole forces.** Reduced total magnetic dipole–dipole force acting on particle 1, $F_1/F_0$, in a chain cluster as a function of the number of dipoles $N$. The squares represents the analytical results for hard sphere dipoles. The circles represents the numerical results for soft-spheres using the WCA-potential, as shown in Fig. 1c. Note that the scale $F_0$ depends on the potential, see text for details.

This remaining chain will show further fission events, since the critical velocity increases with an increasing number of particles, see again Fig. 1c.

**Spontaneous fission of a ring**. Now we consider the dynamical behaviour of an isolated ring-like cluster for different numbers of microswimmers $N = 3$, 4 and 5 with vanishing total magnetic moment $\mathbf{M} = 0$ (refs 9,25,40). The activity of each particle leads to the formation of a cluster, which rotates with a constant angular velocity $\omega$ around the non-moving center of mass. The angular velocity depends on the magnitude of the self-propulsion velocity[37].

To study the fission of such a ring-like cluster, we analyse the self-propulsion velocity dependence of the inverse radius of gyration $R_g$ for the final configuration of the cluster, with respect to the initial radius of gyration $R_g(0)$ for a passive cluster, see Fig. 4. The radius of gyration $R_g$ is defined by

$$R_g^2 = \frac{1}{N}\sum_{i=1}^{N}(\mathbf{r}_i - \mathbf{r}_c)^2, \tag{2}$$

where $\mathbf{r}_c = \frac{1}{N}\sum_{j=1}^{N}\mathbf{r}_j$ is the center of the cluster. If each particle is attributed a fictive unit mass, this center can be interpreted as the center of mass. However, in the case of low Reynolds numbers, it is more appropriate to interpret it as the center of velocity. While $R_g(0)/R_g > 1$ indicates shrinking, $R_g(0)/R_g < 1$ implies swelling of the ring-like cluster. In the extreme case of fission, the inverse radius of gyration vanishes, $R_g(0)/R_g \rightarrow 0$.

Figure 4 shows that for ring-like clusters with $N = 3$, 4, 5 magnetic microswimmers all swimmer types show a diverging radius of gyration, so fission, for large self-propulsion velocity. Previous studies without HIs have shown that ring-like clusters are stable for all self-propulsion velocities[37]. However, the critical velocity to achieve fission as well as the general behaviour of the inverse radius of gyration strongly depends on the swimmer type and on the respective number of swimmers in the cluster. The fission of rings is a combined result of a swelling of the ring-like structure and a change in the orientation of the dipoles within the rings—with increasing velocity they point outwards, an effect which is enhanced by HIs.

Let us start with a ring-like cluster of $N = 3$ microswimmers. Figure 5a shows the flow field of a single microswimmer and indicates the position of the two other particles within the ring by grey dots. While these additional particles are attracted in case of pushers, the neighbouring particles are repelled in case of neutral swimmers and pullers. Hence the rings show the expected shrinking and swelling behaviours, see again Fig. 4a. The velocity dependence of the inverse radius of gyration for pushers is

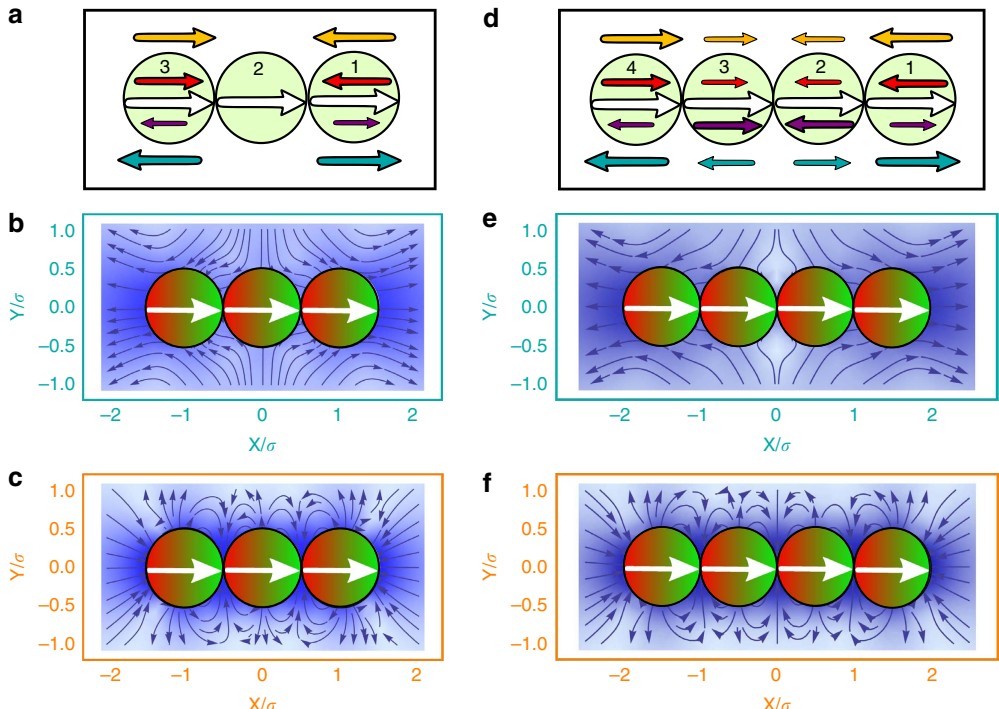

**Figure 3 | Acting forces and flow field around chains.** (**a**,**d**) Acting forces in a chain of $N = 3$ and $N = 4$ microswimmers. The arrows indicate direction and magnitude of all acting forces: self-propulsion force (white), force due to magnetic dipole–dipole interaction (red) and the basic HI due to the Stokeslet (purple). In case of pushers, the extensile flow leads to an repulsion (cyan) and in case of pullers the contractile flow leads to an attraction (orange arrows). Resulting flow field for a linear chain of (**b**) $N = 3$ and (**e**) $N = 4$ pushers as well as for (**c**) $N = 3$ and (**f**) $N = 4$ pullers using the same colour coding as in Fig. 1b.

non-mononotic. This is a result of an induced torque on each particle, Fig. 6a, which leads to a ring configuration where the dipoles point outwards. This state is non-favorable for magnetic dipoles and, therefore, even pushers show a swelling behaviour for velocities $v/v_0 > 2.7$.

Ring-like clusters of $N = 4$, 5 pushers and pullers show the opposite shrinking/swelling behaviour as rings of $N = 3$ particles, see Fig. 5b,c. For pushers, the position of the nearest neighbour are now in the repulsive regime of the flow field, while the nearest neighbours of a pusher are attracted. As before the ring contracting for small self-propulsion velocities, shows a non-monotonic inverse radius of gyration as a function of velocity, see Fig. 4b,c, caused by the weaker magnetic attraction of the misaligned dipoles, see Fig. 6b,c.

The fission velocity for rings of pullers and neutral swimmers is monotonically increasing as a function of number of swimmers in the ring, while it is non-monotonic for pushers, see Supplementary Fig. 1 for the fission state diagram for rings.

**Induced fission and fusion**. Let us now study the fusion scenarios of a ring-like cluster if it collides with a single swimmer. Initially, this swimmer approaches the center of velocity of the ring, here we choose a ring of $N = 5$ swimmers, whereby all swimmers have the same self-propulsion velocity, see Fig. 7a. As long as the ring-like cluster does not show any spontaneous fission the initial set-up leads to an induced collision. In Fig. 7b, we summarize the resulting states for pushers, neutral swimmers and pullers and varied the self-propulsion velocity for these collision events.

As seen before large self-propulsion velocities will lead to a spontaneous fission of the ring-like clusters for all swimmers.

If the initial ring-like cluster is stable and the single swimmer collides with the rotating ring, two different scenarios can be observed. Either the single swimmer fuses with the ring, creating a new stable ring of $N + 1$ particles (see Supplementary Movie 3), or the impact leads to an induced fission into various fragments of different size and shape. While the induced fusion emerges for all considered types of microswimmers, the induced fission does not occur for pushers due to the instability of the initial ring beyond a self-propulsion of $v/v_0 > 0.3$.

**Spontaneous fusion of two ring-like clusters**. Spontaneous pattern formation and pair interaction of rotating particles and vortex arrays, has been aroused a lot of interest in the past few years, in experiments[39–41] and in theory[42–45] revealing the importance of the HI in this type of systems. It has been proved that magnetic rotors at finite Reynolds number exhibit pattern formation and hydrodynamic repulsion, while a pair of interacting rotors in the Stokes limit exhibit hydrodynamic attraction.

In this section, we will consider a pair of rings A, B with $N = 5$ particles with an initial distance $d_{AB}$. Due to the self-propulsion of the individual swimmers the rings can be treated as active rotors[42]. While the center of velocity for an isolated ring is non-motile, a pair of rotors/rings interact via the long-range HI leading to a cooperative motion of their centers of velocity. The actual characteristics of this motion depends on the vorticity of the ring-like cluster: For opposite vorticities the HI leads to a linear translation of the pairs, while for the same vorticity the rotors propagate on a spiral[42–44].

Let us start by considering rotors with the same vorticity. Ring-like clusters composed of neutral swimmers approach each other on bended lines, see dashed lines in Fig. 8a. For large

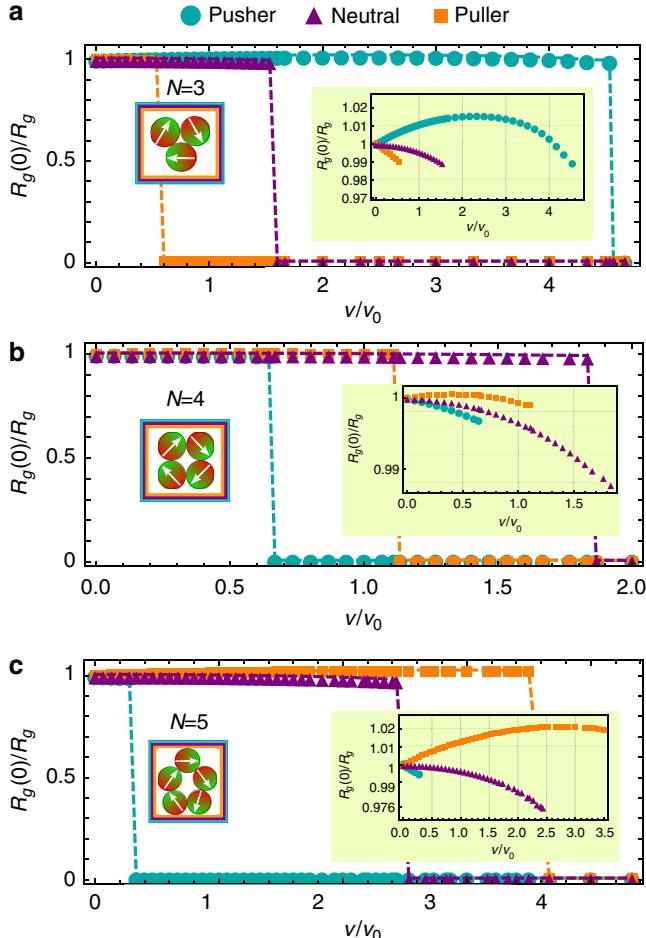

**Figure 4 | Stability of microswimmer rings.** Inverse radius of gyration $R_g(0)/R_g$ as a function of self-propulsion velocity $v$ for ring-like clusters with a given number of swimmers: (**a**) $N=3$, (**b**) $N=4$ and (**c**) $N=5$. Insets: (right) close up the the radius of gyration and (left) sketches of the initial configurations.

self-propelled velocities $v/v_0 > 0.5$ the spiral trajectories, in the case of pullers and pushers, correspond to predictions for active rotors with higher-orders in the mutipole expansion[42]. The centers of velocity of the individual rings propagate on shrinking (puller) or expanding (pusher) spirals, see Fig. 8a. In this case, we have the contribution of the two mobility terms in the motion equation (see equation (5)), that leads to a radial (Stokeslet) and an azimuthal (rotlet) contribution in the cycled-averaged flow field for large distances $d_{AB} \gg R_g$. Here it is important to notice that in our case we have a non-zero torque on each particle due to the dipole–dipole interactions, this makes our problem different from the past studied cases[42,43].

As a result, ring-like clusters of pullers or neutral swimmers will approach each other and collide, while rings of pullers would repel each other for self-propulsion velocities $v/v_0 > 0.5$. However, this velocity is beyond the fission velocity of a single ring, therefore, we never observed this behaviour for rings of $N=5$ pusher. Note that ring-like clusters composed of $N=3$ and $N=4$ pushers have shown the expected propagation[42,43] for self-propulsion velocities $v/v_0 > 0.5$. For the velocities that allows stable rings, $v/v_0 < 0.3$, they behave similar to neutral swimmers. This threshold, $v/v_0 \sim 0.5$, is determined by the velocity dependence of the contractile flow profile.

Figure 9 shows the emerging states after the encounter between the two rings for all microswimmer types. As expected from the center of velocity trajectories, we observe fusion scenarios for small self-propulsion velocities for all swimmers. For pushers in the velocity range $0 \le v/v_0 \le 0.27$, the two rings will collide and stick together but not change their shape—this conformation will be referred to as a dimer (Supplementary Movie 4). Due to the vorticity of the individual rings the dimer will still rotate. The same structure can be found for neutral swimmers ($0 \le v/v_0 \le 0.67$) and pullers ($0 \le v/v_0 \le 0.5$). As shown before large self-propulsion velocities destabilize the rings and spontaneous fission is observed. However, neutral swimmers and pullers exhibit even more distinct emerging states. While the neutral swimmers just show a collision induced fission for self-propulsion velocities $0.67 < v/v_0 \le 2.5$, pullers exhibit two different fusion states. For self-propulsion velocities $0.5 < v/v_0 \le 0.8$, the two rings fuse into a single ring with $N=10$ particles (Supplementary Movie 5). As a last state, we find a state with two counter-rotating microswimmers close to a ring of $N=8$ pullers for $0.8 < v/v_0 \le 1$. These two counter rotating microswimmers are trapped by the superposition of hydrodynamic and dipole–dipole interactions (Supplementary Movie 6).

Now, we focus on rotors with opposite vorticity. For opposite vorticity pullers and neutral swimmers show converging lines in the full range of velocities. While we would find diverging lines for pushers for $v/v_0 > 0.5$, we could not observe this due to the spontaneous fission above $v/v_0 \ge 0.27$ for rings of $N=5$ pushers. Ring-like clusters composed of $N=3$ and $N=4$ pushers have shown the expected propagation[42,43] for self-propulsion velocities $v/v_0 > 0.5$. For velocities that guarantee stable rings of pushers behave similar to rings of neutral swimmers and approach each other on converging lines, see Fig. 8b. For the corresponding velocity regime the contribution due to the mobility terms are larger, in the far-field, compared with the activity terms in the equation of motion (see equation (10)). Therefore, clusters of all swimmers types will collide even if they have opposite vorticities.

For pushers in the regime $0 \le v/v_0 \le 0.2$, we observed an emerging coupled dimer, see Fig. 10 (Supplementary Movie 7). Here the two rings are coupled and translate together on a linear path in the absence of any global rotation. For larger velocities, $0.2 < v/v_0 < 0.3$ the rings can exchange particles and moves with a complex combination of toddling and rotation (Supplementary Movie 8). Neutrals swimmers exhibit the same two states, coupled dimer for $v/v_0 \le 0.33$ and the toddling and rotation state for $0.33 < v/v_0 \le 0.5$. In addition to that the fusion into one ring containing all particles is possible for $0.5 < v/v_0 \le 0.67$. Below the critical velocity for the spontaneous fission of an individual ring, the neutral swimmers exhibit some collision induced fission into various fragments of different size and shape. In contrast to pushers and neutral swimmers, the collision of two ring-like clusters of pullers show the fusion into a single ring and the induced fission as well show a new conformation. The new state, $0 < v/v_0 \le 0.5$, shows a ring of $N=8$ pushers and a chain of two swimmers orbiting around the ring against its direction of rotation (Supplementary Movie 9).

**Discussion**

In conclusion, we have introduced the concepts of spontaneous and induced fission and fusion for clusters of magnetic microswimmers and exemplified these processes by studying linear chains and rings upon increasing the self-propulsion as well as ring–ring collisions. A wealth of dynamical processes governed by the excluded volume, magnetic dipole interactions and by the swimmer type, that is, whether the particles are pushers or pullers, were obtained. The type of swimming even dictates the qualitative scenario. As an example, linear magnetic chains do not show fission for pullers but do so for pushers.

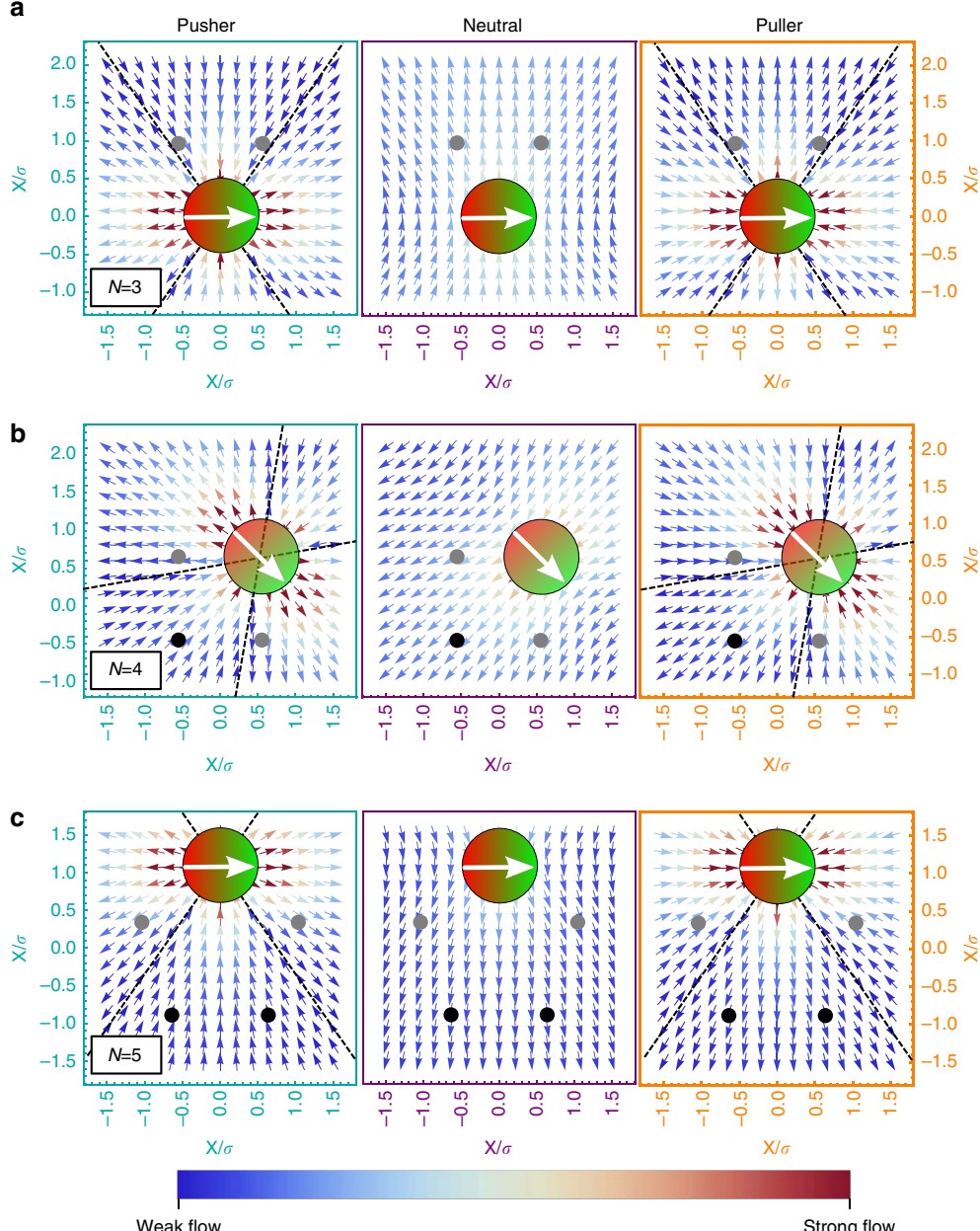

**Figure 5 | Flow-field of a single swimmer within a ring-like cluster.** Flow field generated by a single (left) pusher, (middle) neutral swimmer and (right) puller. Dots indicate the positions of nearest (grey dots) and next nearest neighbours (black dots) for (**a**) a ring-like cluster of $N = 3$, (**b**) $N = 4$ and (**c**) $N = 5$. The dashed lines indicate the transition from inwards to outward flow. The magnitude of the flow is indicated by the colour code varying from dark blue (weak flow) to dark red (strong flow).

A ring-like swimmer cluster which is stable for self-propelled particles without HIs[37] shows fractionation both for pushers and pullers. This demonstrates how a complex active swimmer can be composed of individual entities generalizing recent work where the active spinners were form-stable[46–53]. Two fusing magnetic ring clusters exhibit complex joint dynamical modes including a coupled dimer motion and a combination of toddling and rotation. This complex self-organization can be used to obtain final translational speed out of an initial pure rotational motion.

Very recent experiments have analysed the motion of self-assembled clusters for active Janus particles. For magnetic dipoles it has been shown that the actuation by an oscillating external magnetic field enables the propulsion of structurally non-deforming chains and rings up to $N = 5$ particles. While this study focused on single clusters, another recent study of non-

magnetic Janus spheres showed that assembled pinwheels can perform a robust propulsion and that two pinwheels with the same chirality synchronize their motion[25]. However, up to now the main focus in experiments using artificial swimmers, was to enable the propulsion of individual clusters without changing their structure[54,55]. Our study provides ideas about how this cluster structure can be tuned systematically. Moreover, in case of magnetotactic bacteria the main recent research goal is the steering of these swimmers by an external magnetic field[56–58]. The recent experimental developments provide evidence that our predictions about fission and fusion, can be verified in granular and colloidal magnetic swimmers[14,25] as well as in magnetotactic bacteria[56–58].

For the future, it would be interesting to put the magnetic microswimmer clusters into further external fields such as an

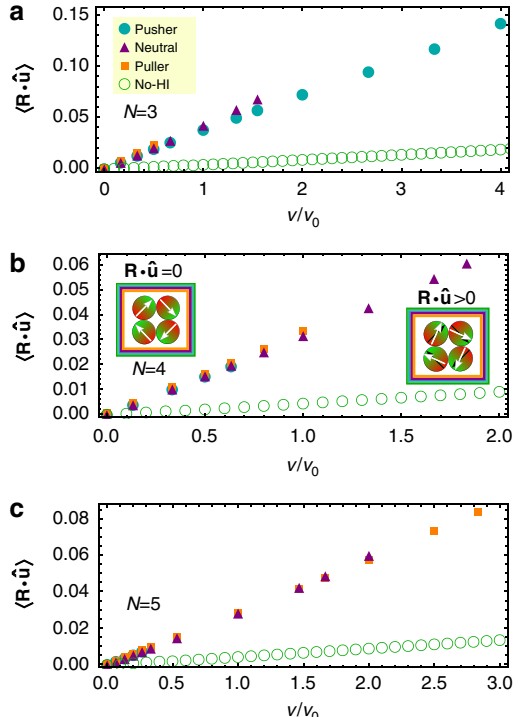

**Figure 6 | Alignment of dipoles within a ring-like structure.** Alignment of the swimmer within a ring-like chain measured by the product of the swimmers orientation $\hat{\mathbf{u}}$ and $\mathbf{R} = \mathbf{r}_i - \mathbf{r}_{cv}$ for (**a**) $N = 3$, (**b**) $N = 4$ and (**c**) $N = 5$ swimmers.

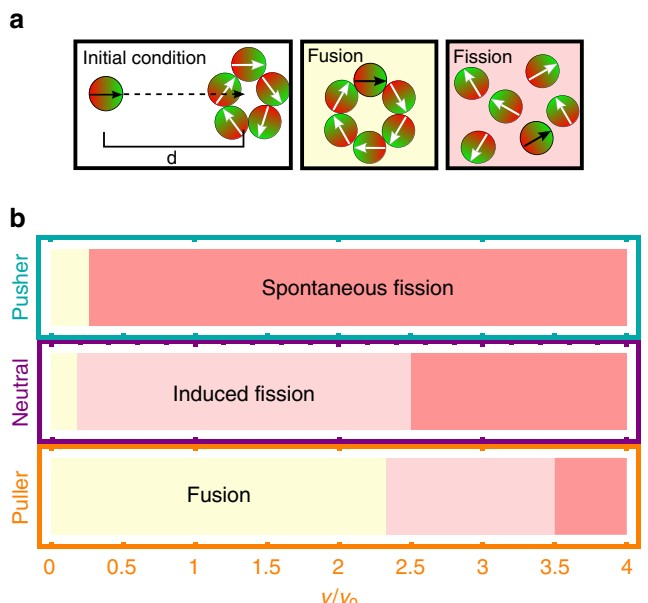

**Figure 7 | Induced fission and fusion.** (**b**) State diagram of the emerging scenarios for all three swimmer types and varied reduced self-propulsion velocity $v/v_0$ and respective snapshots, if a single swimmer collides with a ring-like cluster, see also Supplementary Movie 3. The initial set-up as well as the emerging characteristic structures are sketched in panel **a**.

external shear flow or an aligning external magnetic field[14,59]. We expect that the fission and fusion processes will be tunable by external fields, which would facilitate an exploitation of our modes for various applications. These include drug delivery by magnetic clusters, assisted by self-propulsion and guided by

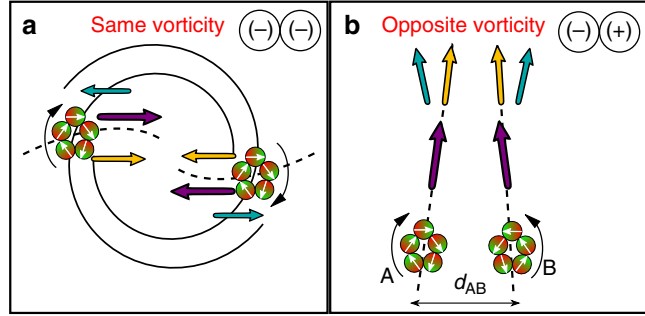

**Figure 8 | Schematic trajectories for two rotating ring-like clusters.** (**a**) Two ring-like clusters A, B of $N = 5$ swimmers each, rotating with same vorticity and an initial distance will approach while propagating on a spiral, while pushers may repel each other. Dashed lines show the trajectories of the individual centers of velocity for neutral swimmers. The coloured arrows indicate the direction of the acting forces due to hydrodynamic interactions, as in Fig. 3. (**b**) For rings rotating with opposite vorticity, the clusters will approach each other for all microswimmer types. Their respective centers of velocity move on straight lines. For pushers it is possible that the rings move apart from each other at larger velocities.

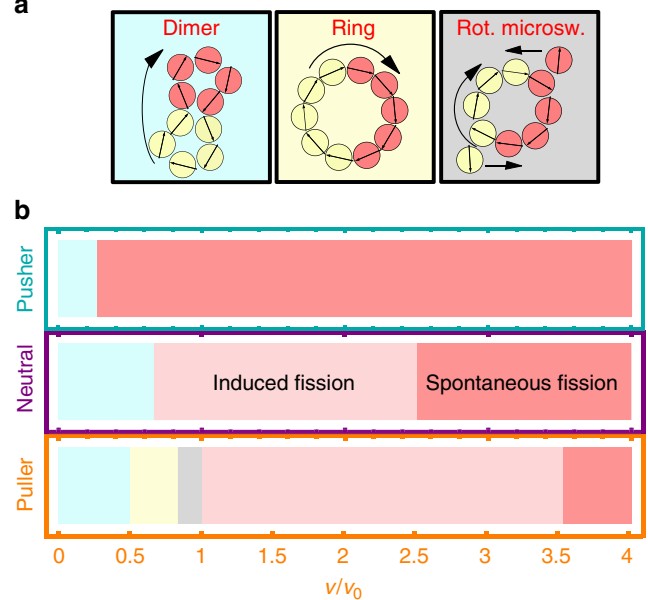

**Figure 9 | Spontaneous fusion and fission for ring-like clusters with the same vorticity.** (**b**) Emerging states for all the swimmers types as a function of reduced self-propulsion velocity $v/v_0$—see Supplementary Movies 4–6. Different states are indicated by different colours and fused states are sketched in panel **a** and characterized as: (i) a spontaneously fused rotating dimer, (ii) an emerging closed ring of $N = 10$ swimmers, (iii) a closed ring of $N = 8$ swimmers with counter rotating microswimmers, (iv) induced fission into various fragments and (v) spontaneous fission of the individual initial ring-like clusters.

external magnetic fields, where the dynamical modes can help to surmount specific barriers and overcome geometric constrictions. Our results can also help to develop novel magnetorheological fluids whose viscoelastic behaviour can be steered by activity and by magnetic fields[60]. In fact, the cluster size of the aggregates largely determines the shear viscosity and the magnetization of the fluid such that combined smart magneto-viscous material properties can result by changing the clustering kinetics. Interestingly, it was recently shown that active colloids may exhibit an effective negative viscosity[61], which is another marked

**a**

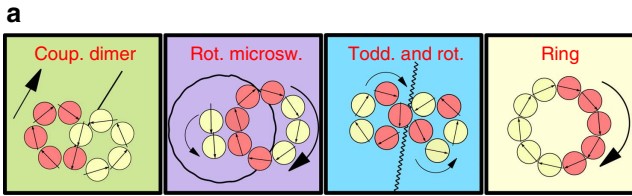

**b**

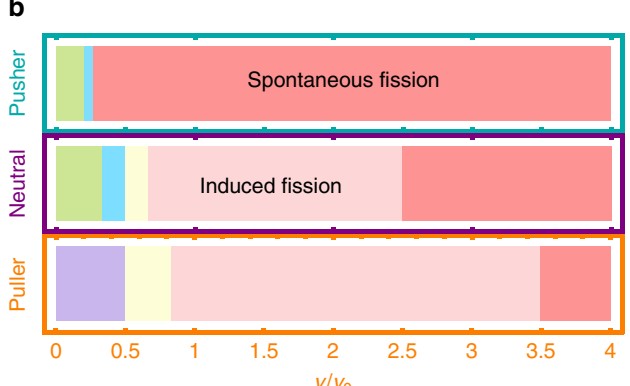

**Figure 10 | Spontaneous fusion and fission for ring-like clusters with opposite vorticity.** (**b**) Emerging states for all swimmer types as a function of reduced self-propulsion velocities $v/v_0$—see Supplementary Movies 7–9. Each state is depicted in a sketch in panel **a** and can characterized by: (i) a non-rotating but translating coupled dimer, (ii) a counter rotating microswimmer chain orbiting around a ring of $N = 8$ swimmers (iii) toddling and rotation, where the former rings exchange swimmers and translate, (iv) an emerging ring containing all $N = 10$ swimmers as well as (v) collision induced and (vi) spontaneous fission of the ring-like clusters.

non-equilibrium feature as induced by self-propulsion. Combining this finding with the rich dynamical cluster scenarios found here will open new doors to construct active composite materials with 'intelligent' and unconventional properties.

## Methods

**Model.** We consider $N$ spherical magnetic microswimmers in three spatial dimensions. The position of the $i$th swimmer is given $\mathbf{r}_i = [x_i, y_i, z_i]$ and its self-propulsion velocity $\mathbf{v}_i = v\hat{\mathbf{u}}_i$ is directed along the unit vector $\hat{\mathbf{u}}_i = [\sin\theta_i \cos\phi_i, \sin\theta_i \sin\phi_i, \cos\theta_i]$. A magnetic dipole moment $\mathbf{m}_i = m\hat{\mathbf{u}}_i$ is directed along the same axis. We model each swimmer as a soft core dipole using a WCA $U^{\text{WCA}}$ and a point dipole interaction potential $U^{\text{D}}$.

$$U_{ij}^{\text{WCA}} = \begin{cases} 4\epsilon\left[\left(\frac{\sigma}{r_{ij}}\right)^{12} - \left(\frac{\sigma}{r_{ij}}\right)^6\right] + \epsilon, & r \le 2^{1/6}\sigma, \\ 0, & r > 2^{1/6}\sigma, \end{cases} \quad (3)$$

$$U_{ij}^{\text{D}} = \frac{m^2}{r_{ij}^3}\left[\hat{\mathbf{u}}_i \cdot \hat{\mathbf{u}}_j - \frac{3(\hat{\mathbf{u}}_i \cdot \mathbf{r}_{ij})(\hat{\mathbf{u}}_j \cdot \mathbf{r}_{ij})}{r_{ij}^2}\right], \quad (4)$$

where $\mathbf{r}_{ij} = \mathbf{r}_j - \mathbf{r}_i$ is the position of swimmer $j$ relative to particle $i$ and $r_{ij}$ their respective distance. The particles are moving in a viscous fluid restricted to the low Reynolds number regime. Each particle creates a long-range flow that disturbs the flow field around the other particles. These interactions are the so-called HIs, which couples the translational and rotational dynamics between microswimmers.

We describe the flow field around each miroswimmer as a linear combination of fundamental solutions of the Stokes equations[62]. We split these HI in two contributions: The first one is related to the mobility which is controlled by the dipole–dipole interactions, these interactions create a Stokeslet and a rotlet in the fluid where the particle is placed. Then, the $k$-component of the velocity field created for the $i$th microswimmer due to the presence of a microswimmer $j$ is,

$$v_k^{\text{mobility}}(\mathbf{r}_i) = \sum_{i\ne j, j=1}^N \mathbf{G}_{kl}(\mathbf{r}_{ij})f_l^j + \sum_{i\ne j, j=1}^N \left(\frac{\mathbf{r}_{ij}}{8\pi\eta r_{ij}^3}\times\mathbf{T}_j\right)_k \quad (5)$$

where $\mathbf{G}_{kl}(\mathbf{r}_{ij}) = \frac{1}{8\pi\eta r_{ij}}\left(\delta_{kl} + \frac{r_{ij,k}r_{ij,l}}{r_{ij}^2}\right)$ is the Oseen tensor, $\mathbf{f}^j = -\nabla_{\mathbf{r}_j}U^{\text{D}}$ is the force and $\mathbf{T}_j = \hat{\mathbf{u}}_j(t)\times\nabla_{\hat{\mathbf{u}}_j}U^{\text{D}}$ is the torque on $j$th particle. The simplest swimmer

considered here is the neutral microswimmer which is basically an active particle[37] that only interact hydrodynamically via the mobility tensor.

The second contribution due to HI is related to the activity and takes account the fluid flow that the microswimmer produces while moving in the fluid. To keep the system as simple as possible, we add a force dipole describing the far-field distortions and a source quadrupole to describe the nearer-field distortions to the equations of motion. The $k$-component of the velocity field created for the $i$th microswimmer is given by

$$v_k^{\text{activity}}(\mathbf{r}_i) = \sum_{i\ne j, j=1}^N \left(\mathbf{G}_{kl,p}(\mathbf{r}_{ij})\mathbf{d}_{pl}^j + \mathbf{Q}_{kl,pt}(\mathbf{r}_{ij})\mathbf{q}_{pl}^j\right), \quad (6)$$

where

$$\mathbf{G}_{kl,p}(\mathbf{r}) = -\frac{1}{8\pi\eta}\left(\left(\delta_{kl}r_{ij,p} - \delta_{kp}r_{ij,l} - \delta_{pl}r_{ij,k}\right) + 3\frac{r_{ij,k}r_{ij,l}r_{ij,p}}{r_{ij}^2}\right), \quad (7)$$

$$\mathbf{Q}_{kl,p}(\mathbf{r}) = \frac{1}{8\pi\eta}\left(-\frac{3}{r_{ij}^5}\left(\delta_{kl}r_{ij,p} + \delta_{kp}r_{ij,l} + \delta_{lp}r_{ij,k}\right) + 15\frac{r_{ij,k}r_{ij,l}r_{ij,p}}{r_{ij}^7}\right), \quad (8)$$

are the derivatives of the Oseen tensor[62,63]. In equation (6), from left to right, the first term corresponds to a point force dipole, a second order tensor which takes into account the self-propulsion with $\mathbf{d}_{pl} = \sigma_0 \hat{\mathbf{u}}_p \hat{\mathbf{u}}_l$ and $\sigma_0 = 3\pi\eta\sigma^2 v$ is the hydrodynamic dipole strength where $\eta$ is the viscosity and $v$ the self-propulsion velocity. Since an isolated microswimmer is force free, we assume that particles are moving due to an effective internal self-propulsion force $\mathbf{f}_0$ oppositely acting to the force generated by the Stokes drag on the surface of the sphere, therefore, $\mathbf{f}_0 = \pm 3\pi\eta\sigma\mathbf{v}$. If this force is anti-parallel to the particle's magnetization $\sigma_0 < 0$ the swimmer is extensile while if $\mathbf{f}_0$ is pointing in the same direction of the magnetization $\sigma_0 > 0$ the microswimmer is contractile. The last term corresponds to a source quadrupole, a second order tensor given by $\mathbf{q}_{pl} = -\sigma_0\sigma^2 \hat{\mathbf{u}}_p \hat{\mathbf{u}}_l/20$ (ref. 62).

Summarizing the HI we have,

$$\mathbf{v}^{\text{HI},i}(\mathbf{r}_i) = \underbrace{\mathbf{v}^{\text{mobility}}(\mathbf{r}_i)}_{\text{neutral}} + \underbrace{\mathbf{v}^{\text{activity}}(\mathbf{r}_i)}_{\text{pusher,puller}}. \quad (9)$$

Microswimmers move in the low Reynolds number regime, therefore the corresponding equations of motion for the positions $\mathbf{r}_i$ and orientations $\hat{\mathbf{u}}_i$ are given by

$$\partial_t\mathbf{r}_i(t) = v\hat{\mathbf{u}}_i(t) + \mathbf{v}_i^{\text{HI}} - \frac{\nabla_{\mathbf{r}_i}U}{3\pi\eta\sigma}, \quad (10)$$

$$\partial_t\hat{\mathbf{u}}_i(t) = \frac{-\mathbf{T}_i\times\hat{\mathbf{u}}_i(t)}{\pi\eta\sigma^3} + \sum_{i\ne j, j=1}^N \left(\frac{\mathbf{r}_{ij}}{r_{ij}^3}\times\frac{\mathbf{f}_j}{8\pi\eta}\right)\times\hat{\mathbf{u}}_i, \quad (11)$$

with $U = U^{\text{D}} + U^{\text{WCA}}$.

**Numerical methods and parameters.** We solve the equations of motion, equations (10 and 11), using the third-order Adams-Bashforth-Moulton predictor-corrector method with a time step $\Delta t = 10^{-4}\tau$. The time is measured in units of $\tau = 3\pi\eta\sigma^3/\epsilon$ with fluid viscosity $\eta$, and particle diameter $\sigma$, which is our length scale and $\epsilon$ the energy scale from the WCA potential. The simulation time is $t_f \sim 5 \times 10^3\tau$ to allow the clusters to achieve new steady-state structures. The initial configurations for the cluster conformations, either chains or rings, are two dimensional and they are generated by a minimization of the potential energy for given $N$ passive spherical dipoles. We vary number of particles for a chain configuration as $N = 2, 3, \ldots, 10$ and use $N = 3, 4, 5$ for ring configurations. For all studies, the magnetic dipole strength $m^2/(\epsilon\sigma^2) = 2$ is fixed and the self-propulsion velocity $v$ is applied instantaneously on each particle. The velocity is normalized by $v_0 = 2m^2/\pi\eta\sigma^5$, the velocity corresponding to the drag velocity caused by the dipole–dipole interactions for a chain of two dipoles in a head-to-tail configuration.

**Data availability.** Data available on request from the authors.

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

## Acknowledgements

This work was supported by the science priority program SPP 1726 of the Deutsche Forschungsgemeinschaft (DFG). A.K. gratefully acknowledges financial support through a Postdoctoral Research Fellowship (KA 4255/1-1) from the DFG. F.G.-L. acknowledges the financial support of Conicyt Postdoctorado-2015 74150045 and helpful discussions with Rodrigo Soto, Adam Wysocki, Arnold Mathijssen and Marco Leoni.

## Author contributions

F.G.-L, A.K., and H.L. designed the research, analysed the data and wrote the paper; F.G.-L. carried out the simulations.

## Additional information

**Competing financial interests:** The authors declare no competing financial interests.

**Publisher's note**: 

**nature COMMUNICATIONS**

DOI: 10.1038/ncomms14298    **OPEN**

# Erratum: Fission and fusion scenarios for magnetic microswimmer clusters

Francisca Guzmán-Lastra, Andreas Kaiser & Hartmut Löwen

*Nature Communications* 7:13519 doi:10.1038/ncomms13519 (2016); Published 22 Nov 2016; Updated 17 Jan 2017

This Article contains an error in Reference 18 that was introduced during the production process. The correct reference should be:

Steinbach, G. *et al.* Stirrers and movers actuated by oscillating fields. Preprint at arXiv:1607.04733 (2016).

Reference 18 should have been cited in the second paragraph of the 'Discussion' section as follows: 'For magnetic dipoles it has been shown that the actuation by an oscillating external magnetic field enables the propulsion of structurally non-deforming chains and rings up to $N=5$ particles[18].' and 'The recent experimental developments provide evidence that our predictions about fission and fusion can be verified in granular and colloidal magnetic swimmers[14,18,25] as well as in magnetotactic bacteria[56–58].'.

