## [Peer Review File · Nature Communications]

Reviewers' comments:

Reviewer #1 (Remarks to the Author):

The article by Guzmán-Lastra et al. describes a theoretical study of the assembly/dis-assembly (fission and fusion) of active particles swimming in a viscous fluid. Three types of particles are considered: pusher, puller and simple swimmers and the simulation scheme includes the hydrodynamic interactions via the Oseen tensor. The novelty with respect to previous works is that the particles carry a permanent moment which forces them to assemble either into chain or rings, the lowest energy configuration of the initial cluster. The idea is nice and the authors describe a lot of different interesting cases where these chains/rings break apart, merge, propel, rotate etc... The article is well written and easy to be understood by the general readership. The referee thus supports publication of this paper, however there are some unclear points/comments which need to be addressed. The aim of these suggestions is to improve the paper trying to make it suitable for a high impact factor journal like Nat. Comm.

- 1) In general, the referee finds the article rather descriptive, where most of the space is invested in illustrating the phenomena observed and in characterizing them. Probably what is missing is a bit more of description of the basic physical processes involved, for example the reasons behind the coherent rupture of the chain during propulsion (Fig.2 bottom), the reason while a pusher or puller will change the dynamic behavior of the clusters. Also a more extensive description on the status of the art and how this manuscript will improve the latter.
- 2) While the idea to refer to fission and fusion may give a look of novelty to the manuscript, the described processes occur on different time and length scales than atomic fission and fusion, and have different interactions involved. Thus more adequate will be to talk change to assembly/disassembly of swimmers.
- 3) The defined parameter in Eq. (1) is very similar to the Reynolds number and could be confused with it.
- 4) Some more words on the fundamental difference between a pusher and a puller will be welcome, and why is so important to consider them.
- 5) Why in Fig.3 the curves are so different?
- 6) In Fig.4 a and d all interactions are described as arrows, thus basically as forces. But HI interactions contain forces and torques which act depending on the relative orientations. Probably a different notation will be clearer.
- 7) Page 3, column 2 the authors talk in terms of center of mass, but here will be more appropriate center of velocity, since we are at low Reynolds number
- 8) The discussion part is basically the conclusions, there are no really discussion in the paper besides description of the phenomena observed. Why these phenomena will be of interest for the community?
- 9) What is the role of thermal fluctuations on the stability of these structures? What happens when the number of particles in the clusters grows? The fission and fusion are no longer symmetric?
- 10) There are more articles on propelling magnetic particles that the authors may want to cite, including few works on magnetotactic bacteria which seems to be the main inspiration point of the

paper.

Reviewer #2 (Remarks to the Author):

The article by Guzmán-Lastra et al. describes fission and fusion processes for active hard dipolar particles in linear and ring like arrangements (rotors) that are known to be the ground states of such systems. Ground states will always be dominant at sufficiently high coupling constants of the dipolar system. These insights were obtained through computer simulations of model systems, including hydrodynamic interactions which have been proven to be important for the study of these non-equilibrium systems. The study of active brownian particles is important for many areas in nano-sciences and biological systems, like sperm and bacteria.

The paper is highly original, and the data and methodology is scientifically sound. The quality of the presentation is extraordinary clear, and the supplemental material in form of the animated simulational data gives additional insight into the behavior of these systems, and makes the core results of the paper more easily accessible.

As far as I can judge the paper has extensively listed the previous achievements of other groups, and has also added sufficiently novelty to the field to grant the acceptance. Microswimmers of this kind really have the potential to become extremely useful tools for many areas in nano-sciences. The paper is written really in a way that also non-specialists can grasp the important ideas. The only minor drawback is the lack of comparisons to experimental data. Maybe the authors could try to make a link to similar systems (if there are any) that have been studied before, or that might be easily studied to verify the predictions of the authors.

REVIEWERS' COMMENTS:

Reviewer #1 (Remarks to the Author):

The authors of the articles have successfully answered to all my queries and they improve the manuscript with their additions such that it now reaches the high level required by Nature Comm. The comparison and the connection with the experimental system is now more clearly explained, and the article can be accepted in Nature Comm.

I would just suggest the authors to add, when they talk in terms of hydrodynamic interactions between assembled magnetic dipoles, [either on page 1, column 2 first red sentences or on page 7, column 2 second red sentences], the following two related references: Phys. Rev. Lett., 115, 138301 (2015); Phys. Rev. Applied, 3, 051003 (2015).

Reviewer #2 (Remarks to the Author):

I am happy with the extensive improvements done by the authors which resulted in a paper that I recommend for publication.

Two minor issues:

1) I am not so certain that I understand the changes made to the definition of Eq. (2), The standard definition would be centre of mass. What exactly is the definition of the center of velocity, and why is it used here. Please clarify, since I am unaware of the relation to the textbook definition of R_G

2) Spelling mistake page 1, col. 2 3. Paragraf "groud" instead of "ground"

Response to First Referee:

The article by Guzmán-Lastra et al. describes a theoretical study of the assembly/dis-assembly (fission and fusion) of active particles swimming in a viscous fluid. Three types of particles are considered: pusher, puller and simple swimmers and the simulation scheme includes the hydrodynamic interactions via the Oseen tensor. The novelty with respect to previous works is that the particles carry a permanent moment which forces them to assemble either into chain or rings, the lowest energy configuration of the initial cluster. The idea is nice and the authors describe a lot of different interesting cases where these chains/rings break apart, merge, propel, rotate etc... The article is well written and easy to be understood by the general readership. The referee thus supports publication of this paper, however there are some unclear points/comments which need to be addressed. The aim of these suggestions is to improve the paper trying to make it suitable for a high impact factor journal like Nat. Comm.

We thank the Reviewer for his/her careful reading of our work and appreciate the suggestions to improve the paper.

Here is a point-to-point reply to the comments of the Referee.

1. In general, the referee finds the article rather descriptive, where most of the space is invested in illustrating the phenomena observed and in characterizing them. Probably what is missing is a bit more of description of the basic physical processes involved, for example the reasons behind the coherent rupture of the chain during propulsion (Fig.2 bottom), the reason while a pusher or puller will change the dynamic behavior of the clusters. Also a more extensive description on the status of the art and how this manuscript will improve the latter.

We thank the Referee for his suggestion to extend the discussions of the presented results in the figures as well as the discussion of all results (along with points 8 and 10 raised by the Referee).

We now already state in the description of the figures on spontaneous fission of a chain that the characteristics of the flow field of the respective swimmer type caused the observed behaviour (see description of Fig. 1). Moreover, we have extended the discussion of Fig. 2(b). The fission process itself is discussed in more detail along with the description of Figs. 3 and 4, showing that the symmetric stepwise fission process can be understood considering the different forces acting within the chain.

We have added the following throughout the text:

Furthermore the impact of hydrodynamic interactions has not yet been studied for magnetic dipole swimmers. [...] The fission process for chains of pushers happens stepwise as can be seen in the time series sketched in Fig. 2(b) and in the Supplementary Movies 1 and 2. The extensile flow field generated by pushers lead to a repulsion of the particles within the chain. Beyond a certain critical velocity, the chain of N particles will split into three different units – a remaining chain of $N - 2$ pushers and repelled head and tail particle. To understand the symmetric stepwise mechanism we will address below the behaviour of the critical velocity leading to fission as well as an analysis of all acting forces within a chain.

2. While the idea to refer to fission and fusion may give a look of novelty to the manuscript, the described processes occur on different time and length scales than atomic fission and fusion, and have different interactions involved. Thus more adequate will be to talk change to assembly/disassembly of swimmers.

The Referee is correct that our observed fission and fusion scenarios are examples of assembly and disassembly. However fission and fusion has been observed in systems of various scales, but not yet in active matter systems. Therefore we kept the term “fission” and “fusion” in the title but have included the terms “disassembly” and “assembly” in the abstract as well as in the introduction.

3. The defined parameter in Eq. (1) is very similar to the Reynolds number and could be confused with it.

We thank the Referee for this remark.

The Reynolds number is usually given by Re . In our manuscript we denote, the end-to-

end distance as R_E , which is a commonly used physical property. Since we used a capital letter E and put it as an index, we think that the current presentation should be clear.

4. Some more words on the fundamental difference between a pusher and a puller will be welcome, and why is so important to consider them.

We thank the Referee for pointing out that we should explain the differentiation of pushers and pullers to the manuscript as well as the reason why it is important to study both cases.

Swimmers move autonomously and are force-free and are divided in two basic classes pushers and pullers. Their far-field hydrodynamics can be well described by a force dipole and pushers and pullers can be distinguished by the sign of this force dipole. While pushers push fluid away from the body along their swimming axis and draw fluid in to the sides, pullers pull fluid in along the swimming direction and repel fluid from the sides. Obviously, this has important consequences for the interactions between swimmers and lead to different fusion scenarios studied in the manuscript.

We have added the mentioned differentiation of pushers and puller in the introduction of the manuscript along with the new Reference 35.

5. Why in Fig. 3 the curves are so different?

The differences of the curves arise due to the softness of the WCA potential. In Figure 3 we study the force on the head particle F_1 of a chain of N magnetic dipoles and compare the results obtained by a soft core (modeled by a WCA potential) and a hard core potential. The force has been calculated for the respective equilibrium structures. In both cases the forces F_1 are normalized by a force $F_0 = 6m^2/\sigma^4$, which is equivalent to the force due to dipole-dipole interaction for two particles in a head-to-tail configuration with a particle distance σ . Hereby σ denotes the length scale of the respectively used potential – either the hard core potential or the WCA potential. While in the case of hard sphere dipoles the interparticle distance $r_{ij} = \sigma$ is fixed, the distance can vary in case of the soft sphere dipoles – here the results have been obtained for a fixed magnetic dipole strength $m^2/(\epsilon\sigma^2) = 2$, where ϵ is the energy scale from the WCA potential.

We have clarified this by extending the discussion of Fig 3.

By considering the total acting magnetic force on the head particle of a chain F_1 we can observe that the magnitude of the magnetic attraction grows with increasing chain length N . Here, we compare the cases of soft magnetic dipolar particles with hard sphere dipoles. In Fig. 3 the force on the head particle F_1 is normalized by $F_0 = 6m^2/\sigma^4$, where σ is the respective length scale provided by the hard core potential or the used Weeks-Chandler-Anderson (WCA) potential [38]. The force F_0 is the force due to the dipole-dipole interaction if a particle pair is in a head-to-tail configuration with a central distance σ . Hence for a larger number of pushers in the chain higher self-propulsion velocities are necessary to split the chain. As a remark, we added these data to Fig. 2(a) as a dashed line, leading again to differences which are caused by the softness of the pair potential.

6. In Fig. 4a and d all interactions are described as arrows, thus basically as forces. But HI interactions contain forces and torques which act depending on the relative orientations. Probably a different notation will be clearer.

In general the Referee is right that HI interactions contain forces and torques. However, in the case shown in Fig 4, the initial set up is a one dimensional chain, with dipoles in a head-to-tail configuration. Due to symmetry there are no resulting torques and this allows us to reduce the plot to the acting forces.

7. Page 3, column 2 the authors talk in term of center of mass, but here will be more appropriate center of velocity, since we are at low Reynolds number

We thank the Referee for this very helpful remark. Indeed it is more appropriate to refer to a center of velocity instead to a center of mass in the manuscript. We have adapted the notation in Eq. (2) and throughout the text.

8. The discussion part is basically the conclusions, there are no really discussion in the paper besides description of the phenomena observed. Why these phenomena will be of interest for the community?

Following the remark of the Referee we have extended our discussion in the manuscript. Along with the recommendation in point 10 we have added a discussion of recent experi-

ments using active Janus particles as well as magnetotactic bacteria. Moreover we discuss now in more detail for which possible future applications our study will be of interest.

Very recent experiments have analyzed the motion of self-assembled clusters for active Janus particles. For magnetic dipoles it has been shown that the actuation by an oscillating external magnetic field enables the propulsion of structurally non-deforming chains and rings up to $N = 5$ particles [19]. While this study focused on single clusters, another recent study of non-magnetic Janus spheres showed that assembled pinwheels can perform a robust propulsion and that two pinwheels with the same chirality synchronize their motion [26]. However, up to now the main focus in experiments using artificial swimmers, was to enable the propulsion of individual clusters without changing their structure. Our study provides ideas about how this cluster structure can be tuned systematically. Moreover, in case of magnetotactic bacteria the main recent research goal is the steering of these swimmers by an external magnetic field [58-60]. The recent experimental developments provide evidence that our predictions about fission and fusion, can be verified in granular and colloidal magnetic swimmers [14,19,26] as well as in magnetotactic bacteria [58-60].

For the future it would be interesting to put the magnetic microswimmer clusters into further external fields such as an external shear flow or an aligning external magnetic field [14,61]. We expect that the fission and fusion processes will be tunable by external fields which would facilitate an exploitation of our modes for various applications. These include drug delivery by magnetic clusters, assisted by self-propulsion and guided by external magnetic fields, where the dynamical modes can help to surmount specific barriers and overcome geometric constrictions. Our results can also help to develop novel magnetorheological fluids whose viscoelastic behaviour can be steered by activity and by magnetic fields [62]. In fact, the cluster size of the aggregates largely determines the shear viscosity and the magnetization of the fluid such that combined smart magneto-viscous material properties can result by changing the clustering kinetics. Interestingly, it was recently shown that active colloids may exhibit an effective negative viscosity [63], which is another marked nonequilibrium feature as induced by self-propulsion. Combining this finding with the rich dynamical cluster scenarios found here will open new doors to construct active composite materials with "intelligent" and unconventional properties.

9. What is the role of thermal fluctuations on the stability of these structures? What happens when the number of particles in the clusters grows?

The fission and fusion are no longer symmetric?

For our studies we used the ground states for soft dipolar spheres as initial configurations. In order to answer the questions about the influence of thermal noise and a larger number of particles we will here differentiate between the fission and fusion scenarios:

(i) Influence of thermal fluctuations on the fission scenarios:

We have shown that an additional self-propulsion of the magnetic dipoles lead to fission of single clusters. Thermal fluctuations will destabilize the cluster configurations even further. Therefore, we expect that thermal fluctuations will decrease the observed critical velocities, which lead to fission of chains or rings, for the all swimmer types.

For chains, the stepwise fission process itself is expected to be affected as well. Thermal noise is expected to show differences from the observed successive repulsion of the head and tail particle. On top of that thermal fluctuations will induce changes in the orientation of the individual swimmers and therefore induce torques which have been prevented in our studies.

(ii) Influence of thermal fluctuations on the fusion scenarios:

The fused ringlike clusters are expected to be stable even in case of thermal fluctuations – however in the states showing *counter-rotating microswimmers* (see Figs. 10 & 11) the orbiting swimmers might be able to move far away from the remaining ring of 8 swimmers.

(iii) Influence of larger number of particles in the fission scenarios:

For chains we have shown that only the pushers will exhibit fission and it will always be a symmetric and stepwise fission process. Increasing the number of particles in the chain will not change this behaviour. According to the fission state diagram for chains of pushers we expect that the fission velocity will monotonically increase if the chains contain more particles. Thanks to the question raised by the Referee we extended our study of the spontaneous fission of rings and show now a fission state diagram for rings in the Supplementary Figure 1. While the fission velocity for ring of pullers and neutral swimmers monotonically increases with increasing number of particles N , the velocity is nonmonotonic for pushers. We already showed that chains are stable for pullers and neutral swimmers – since for increasing number of particles $N \rightarrow \infty$ in a ring-like configuration the radius of curvature vanishes, this is the expected result that for pullers and neutral swimmer a higher velocity is needed to split a ring.

Figure 1: **Fission state diagram for rings.** Emerging state diagram spanned by reduced velocity v/v_0 and number N of particles for ring-like clusters.

For chains of pushers we could successfully explain, that a larger chain leads to a stronger attraction of the head and tail particles and therefore the critical velocity increased with increasing chain length (number of particles within the chain). So we expect the same behaviour for large rings. In fact, the fission velocity decays as a function of number of particles up to $N = 5$ and increases for larger number of particles.

We have added the following remark to the section Spontaneous fission of a ring:
The fission velocity for rings of pullers and neutral swimmers is monotonically increasing as a function of number of swimmers in the ring, while it is non-monotonic for pushers, see Supplementary Figure 1 for the fission state diagram for rings.

(iv) Influence of larger number of particles in the fusion scenarios:

Here we expect that even more scenarios will occur if ring-like clusters with a larger of particles will collide. However, we are confident that the fusion into a single ring-like clusters will still be possible in a respective self-propulsion velocity regime.

10. There are more articles on propelling magnetic particles that the authors may want to cite, including few works on magnetotactic bacteria which seems to be the main inspiration point of the paper.

In the revised manuscript we were able to compare our system to two very recent experiments using Janus particles. In the new Ref. 19 by Steinbach et al. magnetic Janus colloids have been actuated by an oscillating external magnetic field. The main goal of this study was to achieve an alternative actuation concept that enables the propulsion of structurally non-deforming objects. So they successfully showed that their actuation allows the linear propulsion of a chain of $N = 5$ particles as well as the rotation of a stable ring of $N = 3$ and $N = 5$ magnetic Janus colloids. However so far the study only considers a single cluster and does not address any fission scenarios.

In another recent experiment (Ref. 26) by Zhang and Granick active Janus particles self-assembled into pinwheels have been studied. Here it could be shown that such chiral structures can perform a rotational motion even without any deformations of the pinwheel. Moreover the cluster-cluster interaction has been studied considering two pinwheels. While they could not observe any fusion scenarios, they observed the synchronized rotation of two pinwheels with the same chirality.

Experiments with magnetotactic bacteria concentrated so far on the steering of these bacteria using external magnetic fields, see Refs. 58, 59, 60. We now refer to these experiments in the Discussion part of the manuscript, see again point 8.

Moreover, we have added Refs. 20, 21 as further realizations of propelling magnetic particles.

Response to Second Referee:

The article by Guzmán-Lastra et al. describes fission and fusion processes for active hard dipolar particles in linear and ring like arrangements (rotors) that are known to be the ground states of such systems. Ground states will always be dominant at sufficiently high coupling constants of the dipolar system. These insights were obtained through computer simulations of model systems, including hydrodynamic interactions which have been proven to be important for the study of these non-equilibrium systems. The study of active Brownian particles is important for many areas in nano-sciences and biological systems, like sperm and bacteria

The paper is highly original, and the data and methodology is scientifically sound. The quality of the presentation is extraordinarily clear, and the supplemental material in form of the animated simulational data gives additional insight into the behavior of these systems, and makes the core results of the paper more easily accessible.

As far as I can judge the paper has extensively listed the previous achievements of other groups, and has also added sufficiently novelty to the field to grant the acceptance. Microswimmers of this kind really have the potential to become extremely useful tools for many areas in nano-sciences. The paper is written really in a way that also non-specialists can grasp the important ideas. The only minor drawback is the lack of comparisons to experimental data. Maybe the authors could try to make a link to similar systems (if there are any) that have been studied before, or that might be easily studied to verify the predictions of the authors.

We thank the Referee for his/her positive report.

In the revised manuscript we were able to compare our system to two very recent experiments using Janus particles. In the new Ref. 19 by Steinbach et al. magnetic Janus colloids have been actuated by an oscillating external magnetic field. The main goal of this study was to achieve an alternative actuation concept that enables the propulsion of structurally non-deforming objects. So they successfully showed that their actuation allows the linear propulsion of a chain of $N = 5$ particles as well as the rotation of a stable ring of $N = 3$ and $N = 5$ magnetic Janus colloids. However so far the study only

considers a single cluster and does not address any fission scenarios.

In another recent experiment (Ref. 26) by Zhang and Granick active Janus particles self-assembled into pinwheels have been studied. Here it could be shown that such chiral structures can perform a rotational motion even without any deformations of the pinwheel. Moreover the cluster-cluster interaction has been studied considering two pinwheels. While they could not observe any fusion scenarios, they observed the synchronized rotation of two pinwheels with the same chirality.

Experiments with magnetotactic bacteria concentrated so far on the steering of these bacteria using external magnetic fields, see Refs. 58, 59, 60.

We now refer to these experiments in the Discussion part of the manuscript.

Very recent experiments have analyzed the motion of self-assembled clusters for active Janus particles. For magnetic dipoles it has been shown that the actuation by an oscillating external magnetic field enables the propulsion of structurally non-deforming chains and rings up to $N = 5$ particles [19]. While this study focused on single clusters, another recent study of non-magnetic Janus spheres showed that assembled pinwheels can perform a robust propulsion and that two pinwheels with the same chirality synchronize their motion [26]. However, up to now the main focus in experiments using artificial swimmers, was to enable the propulsion of individual clusters without changing their structure. Our study provides ideas about how this cluster structure can be tuned systematically. Moreover, in case of magnetotactic bacteria the main recent research goal is the steering of these swimmers by an external magnetic field [58-60]. The recent experimental developments provide evidence that our predictions about fission and fusion, can be verified in granular and colloidal magnetic swimmers [14,19,26] as well as in magnetotactic bacteria [58-60].

Response to First Referee:

The authors of the article have successfully answered to all my queries and they improve the manuscript with their additions such that it now reaches the high level required by Nature Comm. The comparison and the connection with the experimental system is now more clearly explained, and the article can be accepted in Nature Comm. I would just suggest the authors to add, when they talk in terms of hydrodynamic interactions between assembled magnetic dipoles, [either on page 1, column 2 first red sentences or on page 7, column 2 second red sentences], the following two related references: *Phys. Rev. Lett.*, 115, 138301 (2015); *Phys. Rev. Applied*, 3, 051003 (2015).

We are glad that the Referee is pleased with the improvements of the manuscript and recommends its publication. The two mentioned references have been added on page 7 – they are now listed as Refs. 57 and 58.

Response to Second Referee:

I am happy with the extensive improvements done by the authors which resulted in a paper that I recommend for publication.

We thank the Referee for his/her positive recommendation for publication of the manuscript.

Here is a point-to-point reply to the comments of the Referee.

1. I am not so certain that I understand the changes made to the definition of Eq. (2). The standard definition would be center of mass. What exactly is the definition of the center of velocity, and why is it used here. Please clarify, since I am unaware of the relation to the textbook definition of R_G .

The Referee is right that the standard definition of the radius of gyration considers the center of mass. For a system of N identical particles the center of mass is the geometric center. In our manuscript, we consider swimmers at low Reynolds numbers and in this regime the mass has no influence on the system since it is fully over-damped. Therefore

the term mass might be misleading as pointed out by the first referee. We have now given the precise definition of the reference position needed for the definition of the radius of gyration and have discussed its interpretation in the manuscript as follows:

The radius of gyration R_g is defined by

$$R_g^2 = \frac{1}{N} \sum_{i=1}^N (\mathbf{r}_i - \mathbf{r}_c)^2,$$

where $\mathbf{r}_c = \frac{1}{N} \sum_{j=1}^N \mathbf{r}_j$ is the center of the cluster. If each particle is attributed a fictive unit mass, this center can be interpreted as the center of mass. However, in the case of low Reynolds numbers, it is more appropriate to interpret it as the center of velocity.

2. Spelling mistake page 1, col. 2, 3. Paragraph “groud” instead of “ground”.

We thank the Referee for his careful reading. We have corrected the spelling.